# Raman Spectroscopy as a Novel Method for the Characterization of Polydioxanone Medical Stents Biodegradation

**DOI:** 10.3390/ma14185462

**Published:** 2021-09-21

**Authors:** Jan Loskot, Daniel Jezbera, Aleš Bezrouk, Rafael Doležal, Rudolf Andrýs, Vendula Francová, Dominik Miškář, Alena Myslivcová Fučíková

**Affiliations:** 1Department of Physics, University of Hradec Králové, Rokitanského 62, 500 03 Hradec Králové, Czech Republic; jan.loskot@uhk.cz (J.L.); daniel.jezbera@uhk.cz (D.J.); dominik.miskar@uhk.cz (D.M.); 2Department of Medical Biophysics, Faculty of Medicine in Hradec Králové, Charles University, 500 03 Hradec Králové, Czech Republic; 3Department of Chemistry, University of Hradec Králové, Rokitanského 62, 500 03 Hradec Králové, Czech Republic; rafael.dolezal@uhk.cz (R.D.); rudolf.andrys@uhk.cz (R.A.); 4ELLA-CS, s.r.o., Milady Horákové 504/45, 500 06 Hradec Králové, Czech Republic; vendula.francova@ellacs.eu; 5Department of Biology, University of Hradec Králové, Rokitanského 62, 500 03 Hradec Králové, Czech Republic; alena.fucikova@uhk.cz

**Keywords:** biodegradable polymer, polydioxanone, stent, Raman spectrum, hydrolytic degradation, crack growth

## Abstract

Polydioxanone (PPDX), as an FDA approved polymer in tissue engineering, is an important component of some promising medical devices, e.g., biodegradable stents. The hydrolytic degradation of polydioxanone stents plays a key role in the safety and efficacy of treatment. A new fast and convenient method to quantitatively evaluate the hydrolytic degradation of PPDX stent material was developed. PPDX esophageal stents were degraded in phosphate-buffered saline for 24 weeks. For the first time, the changes in Raman spectra during PPDX biodegradation have been investigated here. The level of PPDX hydrolytic degradation was determined from the Raman spectra by calculating the area under the 1732 cm^−1^ peak shoulder. Raman spectroscopy, unlike Fourier transform infrared (FT-IR) spectroscopy, is also sensitive enough to monitor the decrease in the dye content in the stents during the degradation. Observation by a scanning electron microscope showed gradually growing cracks, eventually leading to the stent disintegration. The material crystallinity was increasing during the first 16 weeks, suggesting preferential degradation of the amorphous phase. Our results show a new easy and reliable way to evaluate the progression of PPDX hydrolytic degradation. The proposed approach can be useful for further studies on the behavior of PPDX materials, and for clinical practice.

## 1. Introduction

Medical stents are tubes used to reinforce hollow tubular parts of the body, damaged by a disease or defective for other reasons, and to restore their patency [1]. Originally, non-degradable materials were used for stent production, therefore stents had to be later removed from the body complicatedly. Nowadays, biodegradable materials, either metals or polymers, are used where appropriate. Biodegradable stents decompose in the human body, so no surgery is required to remove them when they are no longer needed [1,2].

For clinical practice, it is essential that the biodegradable stent can support the tissue for a specified period. Reliability plays a key role here, i.e., the stent should disintegrate after a defined period with minimal variability. Detailed knowledge of the degradation process is also desirable for fine-tuning the stent manufacturing process.

Biodegradable and bioabsorbable polymer materials are absorbed in the human body without unfavorable consequences and are eventually excreted. These materials are used for various surgical purposes, such as sutures [3], surgical devices (pins, anchors, clips) [4,5] or tissue engineering [6,7,8,9,10]. Aliphatic polyesters made from lactone monomers or copolymers thereof are popular. Frequently used monomers are, among others, lactide, glycolide, *ε*-caprolactone and *p*-dioxanone [3,11,12,13].

One biodegradable material used for the fabrication of stents is a poly(*p*-dioxanone) (polydioxanone, PPDX) fiber, from which a tube of the desired shape and dimensions is knitted. Such stents have been inserted, e.g., into the esophagus [14,15], intestine [16], bronchus [17], trachea [18] or bile duct [19]. PPDX stents have been widely studied in in vivo medical trials [14,16,19,20,21,22], their mechanical properties have been investigated [23,24,25,26], and their degradation has also been studied to some extent from a physicochemical point of view [25,27]. Stents made of similar materials have also been analyzed in studies [28] (poly(L-lactic acid) endovascular stent) and [29] (intracoronary stent coated with poly(L-lactic acid)).

The manufacturing of absorbable sutures and other surgical devices from PPDX was patented in 1977 [30] and since the early 1980s, this material has been manufactured and widely used in medicine [31]. PPDX is a semi-crystalline aliphatic polyester that is produced by the ring-opening polymerization of *p*-dioxanone. The structure of its monomer unit is –[CH_2_–CH_2_–O–CH_2_–CO–O]_n_–. The great advantage of PPDX lies in its excellent bioabsorbability [32]. PPDX fibers are typically produced by the extrusion method [33,34].

The morphology of poly(*p*-dioxanone) was first studied by Furuhashi et al. [35]. Using electron diffraction on PPDX crystals and X-ray diffraction on PPDX fiber, they determined the crystal structure (orthorhombic P2_1_2_1_2_1_ space group) and the lattice constants (a = 0.970 nm, b = 0.742 nm, c = 0.682 nm (chain axis)). Their conclusions were confirmed and supplemented by a proposal for the molecular conformation by Gesti et al. [31]. Ooi and Cameron [36] determined the period between the lamellae in the fibers as 10 nm using SAXS (Small-angle X-ray Scattering). Jaidann and Brisson [37] determined the orientation distribution of these lamellae using Raman spectroscopy (RS) and X-ray spectroscopy.

For medical use, it is important to understand the behavior of PPDX during its biodegradation. The degradation of polyesters in the body environment generally consists in the cleavage of ester bonds, caused primarily by hydrolysis, ordinarily with the contribution of certain enzymes [31,38]. Sufficiently small oligomers are then eluted out of the material. The degradation process leads to changes in the proportion of the crystalline phase, affecting the mechanical properties of the material too [38,39]. Typical advanced methods for determining the extent of hydrolytic degradation are gel chromatography [40] as well as variants of mass spectrometry [41], including analysis of degradation products [42].

Compared to widely used poly(L-lactic acid) and polyglycolide, PPDX contains a smaller proportion of ester groups, so it degrades more slowly, which is more suitable for many applications. The hydrolytic degradation of PPDX was studied by Márquez et al. [43] and Sabino et al. [44], who compared thin PPDX films before and after the degradation. Ooi et al. [36], Li et al. [27] and Sabino et al. [45] dealt with changes in various physical and mechanical properties of PPDX fibers during hydrolytic degradation. In the latter study, also the effect of degradation on the fiber surface was investigated by a scanning electron microscope (SEM). Li et al. [27] measured and interpreted Fourier transform infrared (FT-IR) spectra of degraded PPDX fibers too. On the contrary, the use of RS to study PPDX degradation has not been addressed yet.

The methods used so far for polymer degradation analysis are either not sensitive enough or too expensive and complicated. Therefore, the main idea of our research was to use RS for this purpose.

RS is a method that allows studying vibrational spectra of various substances, providing more detailed information about the bonds of individual atoms in molecules and the material structure. It can be advantageously used for polymer analysis because it is usually sensitive to such materials. Besides chemical composition, RS allows investigating crystalline properties (including the orientation of crystallites in the material), temperature and mechanical effects on the polymer, and polymer degradation [28,29,37,40,46].

In this study, we applied RS to investigate PPDX stent fibers subjected to degradation in simulated body fluid for a total of 24 weeks. The Raman spectra measurements were supported by SEM and atomic force microscopy (AFM) imaging of the stent fibers enriched with a detailed quantitative description of crack formation on the fiber surface. Further, changes in crystallinity were studied by a differential scanning calorimetry (DSC) and the dependency of FT-IR spectra on the degradation time was investigated.

## 2. Materials and Methods

### 2.1. Degradation Process

Esophageal biodegradable stents (Figure 1) manufactured by ELLA-CS (Hradec Králové, Czech Republic) were used as research samples. These stents are braided from 0.65 mm thick fiber made of poly(*p*-dioxanone) dyed with 1-hydroxy-4-(*p*-tolylamino)anthracene-9,10-dione (also known as Solvent Violet 13, content < 0.1 wt%). The stents are 80 mm long, 25 mm in diameter, with flared ends of 31 mm in diameter. The initial mass of the stent is 3.6 g. According to the manufacturer, the integrity and radial force of the stent is maintained until 6–8 weeks after implantation, and subsequently, the radial force gradually decreases until the complete disintegration of the material after 11–12 weeks [47].

A total of five stents of the same type were examined. One of these stents was not degraded at all, being stored at room temperature in a desiccator to prevent the effect of air humidity on it. The remaining four stents were immersed in phosphate-buffered saline (PBS) of pH 7.4 ± 0.2 at body temperature (37 °C) for 4, 8, 16 and 24 weeks with constant stirring. The volume of PBS per one stent was approximately 300 ml. After pulling out of the PBS, these stents were stored at room temperature in a desiccator too. Samples approximately 1 cm in length (Figure 1b) were cut from the stents for subsequent analyses.

### 2.2. Determination of Crystallinity by Differential Scanning Calorimetry (DSC)

The crystallinity of the stents after each degradation period was studied by the means of differential scanning calorimetry using a DSC823^e^ Measuring Module (Mettler-Toledo, Switzerland). Each sample weighing ~9 mg was placed in 0.1 cm^3^ aluminum crucible and the lid was sealed. The scanning rate was 5 °C min^−1^ and the temperature ranged from −30 °C to 150 °C. The percentage of crystalline phase of the polymer was determined from the enthalpy values of the phase transitions, based on the thermogram measured during the first heating scan. Each measurement was performed three times and the results were averaged.

### 2.3. Surface Morphology

The surface morphology of the stents after each degradation period was observed by a scanning electron microscope FlexSEM 1000 (Hitachi, Japan) at an accelerating voltage of 5 kV. Images of five randomly selected rectangular top surface regions (84 µm × 63 µm) were taken from each stent. The SEM was working in secondary electrons mode and the used magnification was 1500× in all cases.

Based on these images, geometrical characteristics of surface structure disruptions (cracks) were quantitatively evaluated using ImageJ 1.52a (Bethesda, MD, USA) software and MS Excel 2019 version 1808 (Redmond, WA, USA). This evaluation was done only for the 8-, 16-, and 24-weeks degraded stents, because the cracks were observed only on these samples.

The following geometrical characteristics of the surface cracks were quantitatively evaluated: Feret’s diameter, Feret’s angle, area, aspect ratio, circularity and solidity. Feret’s diameter is defined as the longest distance between any two points along the selection (i.e., crack) boundary. It is also known as “maximum caliper”. Feret’s angle is the angle between the positive horizontal axis of the image and the line representing the Feret’s diameter of the selection. It can range from 0 to 180 degrees. Aspect ratio (*AR*) is the ratio of major axis to minor axis of the selection’s fitted ellipse:$$AR=\frac{fitted\;ellipse\;major\;axis}{fitted\;ellipse\;minor\;axis}$$

Circularity (*cr*) is defined as follows:$$cr=4\pi \cdot \frac{area}{perimeter^{2}}$$
having values in the range from 0 (highly elongated selection) to 1 (perfect circle). Solidity (*so*) expresses how much the selection is compact. It can be calculated according to the following formula:$$so=\frac{area}{convex\;area}$$
where the convex area is an area delimited by a string wrapped tightly around the selection’s boundary. All these quantities are described in more detail in the ImageJ manual [48].

In addition to these measurements, fracture surfaces of all the stents were observed. For this purpose, broken pieces of the stents’ fibers were first coated with 6 nm thick platinum layer by an EM ACE200 sputter coater (Leica, Germany).

The SEM observations were supplemented by atomic force microscopy measurements using the NaioAFM device (Nanosurf, Switzerland) working in contact mode. The topography of the stents’ top surfaces was displayed, including cracks induced by the degradation process.

### 2.4. Infrared Spectroscopy

Infrared spectra of the stents after each degradation period were acquired by an FT-IR spectrometer ALPHA II (Bruker, Billerica, MA, USA). To verify the reproducibility of the measurement, three samples of each stent were measured and the results were averaged. The spectral range was 400 to 4000 cm^−1^ and the acquisition time of each spectrum was 70 s. All the stent fibers were positioned with the same orientation during the measurement in order to minimize the influence of sample orientation on the spectra.

Additionally, the FT-IR spectrum of the pure Solvent Violet 13 dye was measured to assess the possible influence of this dye on the above spectra. The same acquisition parameters as before were used, and the measurement was repeated three times to verify its reproducibility and the results were averaged.

### 2.5. Raman Spectroscopy

Raman spectra of the stents after each degradation period were acquired by a Raman micro-spectrometer XploRA PLUS (Horiba, Japan) with an excitation laser at a wavelength of 785 nm, a 1200 g mm^−1^ grating and a 10× objective. On the top surface of each stent, a total of 64 spectra were measured in a regular 8 × 8 grid. The grid points were spaced 12.9 µm from one another in the x direction and 11.4 µm in the y direction. All the stent fibers were oriented in the x direction (parallel to the polarization direction of the excitation laser) during the measurement. The measurement arrangement is shown in Figure 2.

The spectra were taken in the range from 200 to 1800 cm^−1^, the acquisition time of each spectrum was 5 min 38 s. The laser power was set to 10 mW.

Additionally, the Raman spectrum of the pure Solvent Violet 13 dye was measured to distinguish which peaks in the above spectra have their origin in this dye. The same laser wavelength and grating as before was used, and the measurement was repeated five times in different positions to verify its reproducibility.

The acquired Raman spectra were further statistically evaluated using MATLAB R2020a software and NCSS 10 statistical software (2015, NCSS, LLC., Kaysville, UT, USA, ncss.com/software/ncss) to characterize their dependency on the stent degradation time. The statistical evaluation was necessary, because the stent degradation impacted mainly low peaks, which were significantly affected by noise.

To interpret the stent degradation, the shoulder of the 1732 cm^−1^ peak was found to be important. Therefore, the area under this shoulder (between 1736 and 1749 cm^−1^) normalized to the height of the 1732 cm^−1^ peak was calculated for each degradation period. At first, the baseline was removed from each spectrum to allow a relevant quantification of the 1732 cm^−1^ peak height and the adjacent shoulder area. Then the shoulder areas were calculated for all the 64 spectra of the particular stent and the results were averaged. Finally, these mean values were compared among the stents to find how they depend on the degradation time. We used the D’Agostino omnibus test to test the normality of the data distribution. The data from normally distributed populations were then described using the mean and standard deviation of the sample ($\bar{X}$ ± SD) while the other data were described using the median and the first and third quartiles of $\tilde{X}$ (1st Q, 3rd Q). Since the normality of the distribution of the same shoulder area data was rejected, we opted to use the Wilcoxon signed-rank test because it does not presuppose a normal distribution. To adjust for multiple comparisons and keep the familywise α at 0.05, the Bonferroni correction was used. The resulting α for a single comparison was 0.0125.

## 3. Results and Discussion

### 3.1. Determination of Crystallinity by DSC

Using DSC, the crystallinity of all the stents was determined. The results (Table 1) indicate that the percentage of crystalline phase was raising from the beginning of degradation until about the 16th week. This was followed by a decrease in crystallinity back to approx. the “8-weeks” value after 24 weeks of degradation. (All DSC measurement protocols are available in Appendix A.)

A somewhat slower increase in crystallinity during the hydrolysis of PPDX was observed in another study [45], where a PPDX monofilament with a diameter of 1 mm was immersed in PBS (pH = 7.4, *t* = 37 °C). The crystallinity increased from 57.6 to 66.0% during first 10 weeks of degradation. The authors also point out that the increase in crystallinity was directly associated with the rise in the material fragility and the decrease in its tenacity.

On the other hand, a more progressive increase in crystallinity of PPDX is described in [49], where the degradation of PPDX fiber scaffolds obtained by electrospinning was carried out. In this case, the crystallinity degree of PPDX immersed in PBS (pH = 7.4, *t* = 37 °C) increased from about 35 to more than 60% over 65 days.

This comparison suggests that the degradation progression depends on the initial crystallinity of the material (amorphous regions are hydrolyzed more easily compared to crystalline regions [45,49]). But it should be noted here that the degradation can also be substantially affected by the manufacturing process of the fibers [45].

The decrease in crystallinity observed after 24 weeks might be caused by the hydrolysis of the crystalline phase, which follows the initial hydrolysis of the amorphous phase. According to Chu [39], the degree of biodegradable polymers’ crystallinity would be maximum in the moment when amorphous regions are removed and the degradation of crystal lattice is beginning. In our case, it would be around the 16th week of degradation.

### 3.2. Surface Morphology

The surface of the non-degraded stent (Figure 3a) observed by the SEM was smooth and fairly homogenous, without visible damage. On the 4-weeks-degraded stent (Figure 3b), the first structural changes in the form of loosened fibrils appeared. No cracks were found either on the non-degraded nor on the 4-weeks-degraded material. Small cracks perpendicular to the longitudinal axis of the stent were observed sporadically on the 8-weeks-degraded stent (Figure 3c) in addition to the fibrils. Their appearance corresponds to crazes typical of polymer regions with a very localized plastic deformation [50].

After 16 weeks of degradation (Figure 3d), the number and size of the cracks increased significantly and sizable fissures parallel to the longitudinal stent axis also appeared. This is consistent with the study [27], where the first similar parallel fissures were observed after 12 weeks of degradation in PBS. As for the 24-weeks-degraded stent (Figure 3e), the dimensions of the cracks were larger again. All images of the stent surfaces are available in Appendix A.

The AFM measurements of the top surface revealed the material topography at the nanoscale and gave evidence of the top surface roughness. As can be seen in Figure 4, the 24-weeks-degraded surface is significantly disrupted and furrowed by deep cracks.

Statistical distributions of the size and shape descriptors of the cracks are shown in Figure 5, and their mean values and medians are given in Table 2. The increase in Feret’s diameter and area shows a considerable crack growth during the degradation. The values of Feret’s angle are still close to 90°, which means that the cracks grow more or less perpendicularly to the axis of the stent. The significant increase in the cracks’ aspect ratio observed between the 16th and the 24th week of degradation suggests that some of the cracks merged together in a longitudinal direction, thereby forming substantially longer gaps. The gradual decrease in circularity is consistent with the cracks lengthening, its significant drop between the 16th and the 24th week of degradation supports the idea of the cracks merging during this period. The decrease in solidity probably means that the resulting cracks have some bottlenecks. The original data on the geometric characteristics of the cracks are available in Appendix A.

The degradation occured preferentially in the amorphous regions of the polymer because the buffer penetrates there more easily than into the crystalline regions [27,50,51]. During the PPDX degradation, the long polymer chains are gradually hydrolyzed into smaller molecules, causing a decrease in the material tensile strength [27]. Based on the above, it can be expected that the cracks were formed predominantly in the amorphous parts of the stents. The disrupted amorphous phase was being eluted during the degradation, which led to an increase in the proportion of the crystalline phase in the material. This was confirmed by DSC measurements (see Section 3.1). As a result, the material brittleness increased with the degradation (the 24-weeks-degraded stent even crumbled easily in fingers).

Diffusion of the solvent molecules also leads to material swelling [27], wherein the solution molecules occupy positions among the polymer molecules. These are forced apart which leads to an increase in their chain separation and the reduction of secondary intermolecular bonding forces [50]. The swelling-induced tension and disruption of intermolecular bonds in the material may have contributed to the growth of the cracks on the stents’ surface.

Figure 6 shows fracture surfaces of the stents. In non-degraded and 4-weeks-degraded stents, bent fibrils were observed after breaking the sample. The fracture of the 8-weeks-degraded stent is characterized by sharp edges of broken fibrils. After 16 weeks of degradation, striation resembling a fatigue fracture was observed. Surface morphology after 24 weeks of degradation indicates a substantially brittle fracture. Surface flaking can be seen there, which can be attributed to the preferential decomposition of amorphous regions leading to the separation of non-degraded pieces of polymer [52].

### 3.3. Infrared Spectroscopy

Figure 7 shows the averaged FT-IR spectrum of the non-degraded stent compared with the averaged FT-IR spectrum of the stent after 24 weeks of degradation. The peaks at 848 cm^−1^ and at 872 cm^−1^ correspond to C–O–C symmetric stretching vibrations and the peak at 1120 cm^−1^ to C–O–C asymmetric stretching vibrations. The 1430 cm^−1^ peak shows –CH_2_– bending vibrations. The strong peak at 1733 cm^−1^ results from C=O stretching vibrations in the ester carbonyl group. The region between 2820 and 3000 cm^−1^ exhibits both symmetric and asymmetric vibrations of –CH_2_– in the aliphatic chain [27,53].

In the region between 1500 and 1650 cm^−1^ a new, low band with a maximum at 1605 cm^−1^ emerged after 24 weeks. It might have been caused by asymmetric stretching of C$\dddot{-}$O in –COO^−^ anions [53] in polymer residues, which were formed as a result of polymer chains scission by hydrolysis of ester bonds. This is consistent with the results of Li et al. in [27], where a change in this spectral region during PPDX degradation is also described and associated with an increase in acidity.

However, by comparing the spectra of the stents with the spectrum of the pure dye (available in Appendix A), it seems that this spectral region may be somewhat affected by the signal coming from the dye (tiny peaks at 1516 and 1586 cm^−1^ correspond to relatively strong peaks of the dye). Therefore, it was not possible to draw more confident conclusions about this region.

The fingerprint region (below 1500 cm^−1^) looks almost the same in the spectra of all the five stents. It is virtually unaffected by the signal coming from the dye, because this signal is significantly weaker than that of the stents. In general, from the mutual similarity of the FT-IR spectra of the stents, it can be concluded that no functional groups were formed to a significant extent during the degradation, except for the mentioned carbonyl anions. The original data for all FT-IR spectra of the stents are available in Appendix A.

### 3.4. Raman Spectroscopy

Figure 8 shows the averaged Raman spectrum of the non-degraded stent compared with the averaged Raman spectrum of the stent after 24 weeks of degradation. The strongest peak at 870 cm^−1^ corresponds to C–O–C symmetric stretching vibration, and the peak at 1048 cm^−1^ results from the stretching vibration of C–C in the aliphatic chain. The 1451 cm^−1^ peak may be caused by –CH_2_– bending. The peak at 1732 cm^−1^ is attributed to C=O stretching vibrations in the ester carbonyl group, as in the case of the FT-IR spectrum [37], [53]. The original data for all Raman spectra of the stents are available in Appendix A.

By comparing the spectra of the stents with the spectrum of the pure dye (available in Appendix A) the following peaks were attributed to the dye: 483, 1242, 1403, 1610, and 1638 cm^−1^. It can be seen in Figure 8 that these peaks were significantly reduced during the degradation. A possible explanation for this is that the dye had been gradually eluted from the stent.

On the other hand, any peak belonging to PPDX did not change noticeably during the degradation, with the exception of the 1732 cm^−1^ peak shoulder. The lessening of this shoulder with increasing degradation time, quantified by calculating the area under this shoulder, is expressed in Table 3. This change in the stent spectrum is probably a manifestation of the degradation of PPDX itself. All the observed decreases in the peak shoulder area were statistically significant (*p* < 0.001), which demonstrates the high sensitivity and reliability of RS.

A correlation can be seen between the decrease in the proportion of the amorphous phase and the decrease in the area under the 1732 cm^−1^ peak shoulder during the first 16 weeks of degradation (Figure 9). Thus, it seems that the structural changes in PPDX expressed by the reduction in the shoulder are related to the content of the amorphous phase.

Further, according to Schrader [53], the whole region 1680–1820 cm^−1^ corresponds to C=O vibrations. Therefore, we suppose that the sharp peak at 1732 cm^−1^ (Figure 8) corresponds to C=O vibrations in the crystalline phase and the 1732 cm^−1^ peak shoulder is related to C=O vibrations in the amorphous phase of PPDX.

The mechanical durability of the stents declared by the manufacturer is 6–8 weeks, which was confirmed by Bezrouk, et al. [23]. They suggest that the reason for such a long lifetime period is a reinforcement effect, i.e., an increase in radial (therapeutic) force, caused by hydrolysis. Since in our study we observed hydrolytic degradation already within the first 4 weeks, the idea of the initial reinforcement of the material by the hydration process may be correct.

Our study was limited by the number of available stents, so longer degradation periods had to be chosen. Because hydrolytic degradation became apparent in less than 4 weeks, it would be desirable to perform analyses at shorter intervals in future studies, considering the time of clinical application. Because only one stent was used for each RS analysis, the results may be influenced by the manufacturing batch. Therefore, in further research, it would be advisable to use multiple stents of different batches for each degradation time interval.

## 4. Conclusions

The Raman spectra of PPDX stents and their dependence on the biodegradation period have been investigated here for the first time. RS has proven to be a simple, fast, and affordable method for both qualitative and quantitative evaluation of the hydrolytic degradation of PPDX biodegradable material. It was found that RS enables evaluation of the level of PPDX degradation by calculating the area under the 1732 cm^−1^ peak shoulder. Statistical tests confirmed that the calculated values of the peak shoulder area are statistically significant.

The RS results together with DSC suggest that the lessening of this shoulder credibly characterizes the breaking of the ester groups of the amorphous phase, leading to scission of the polymer chains. In follow-up research, it would be helpful to explain theoretically which particular structural transformations of PPDX are reflected in the changes in the shape of this spectral region.

Further, it was found that RS is a sufficiently sensitive method to monitor dye content loss in the stents during the degradation period, even though the declared initial dye concentration was <0.1 wt%. In contrast, FT-IR is not as sensitive to the dye content and therefore cannot be used for this purpose.

Morphological changes of the material were also investigated and the gradual time-dependent growth of cracks on the stent fiber surface was quantitatively evaluated. The material crystallinity was increasing during the first 16 weeks, indicating preferential degradation of the amorphous phase.

Our findings are consistent with the results of previously reported mechanical measurements of these stents and with the lifetime period declared by the manufacturer too. The presented results show a new way to assess the progression of PPDX hydrolytic degradation under simulated physiological conditions. Our proposed approach can find its usage also in further studies on the behavior of PPDX stents in the body environment. Reliable quantitative evaluation of PPDX degradation is essential not only for clinical practice but also for fine-tuning the stent manufacturing process.

## Figures and Tables

**Figure 1 materials-14-05462-f001:**
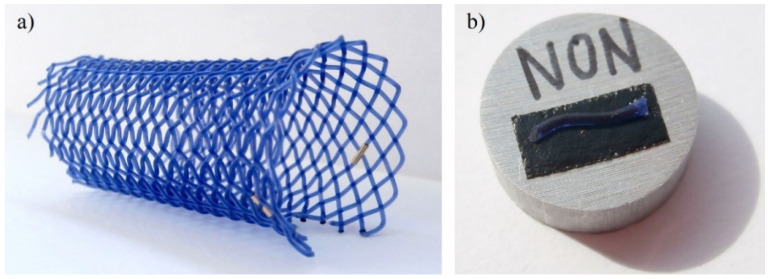
A biodegradable polydioxanone (PPDX) stent sample (**a**) and a piece of its fiber prepared for observations by a scanning electron microscope (SEM) (**b**).

**Figure 2 materials-14-05462-f002:**
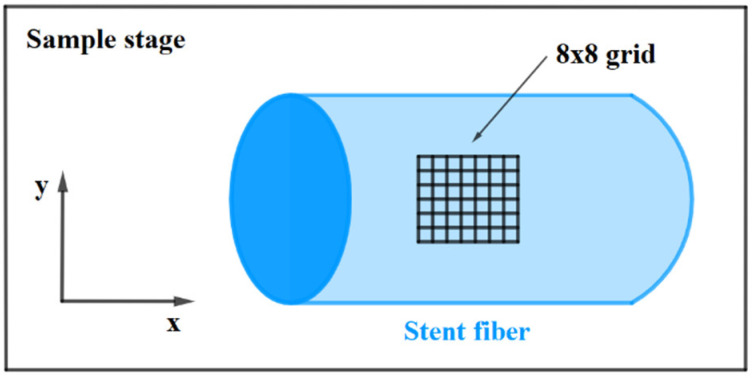
A principal scheme of the Raman measurements. The excitation laser beam impinges on the sample in the z (vertical) direction.

**Figure 3 materials-14-05462-f003:**
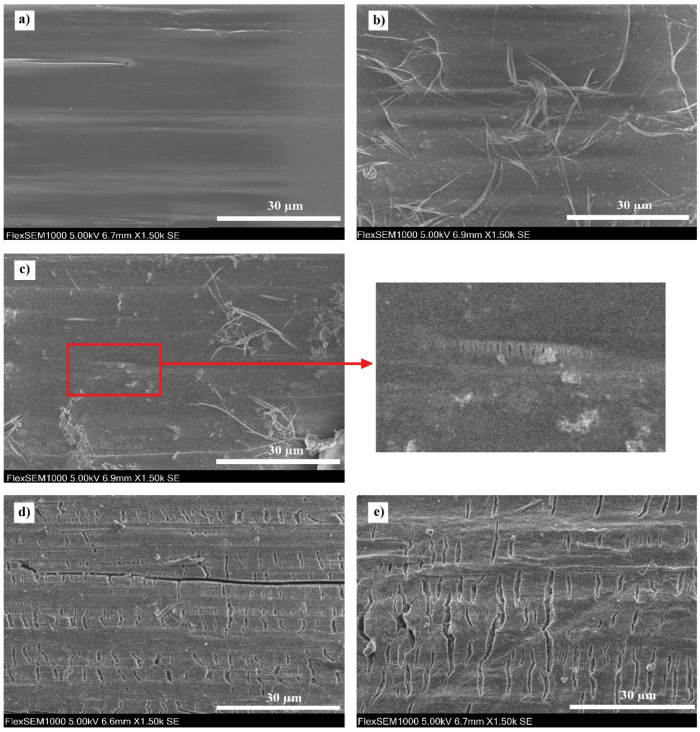
Surface morphology of the stents: (**a**) non-degraded, (**b**) 4-weeks-degraded, (**c**) 8-weeks-degraded, (**d**) 16-weeks-degraded and (**e**) 24-weeks-degraded stent.

**Figure 4 materials-14-05462-f004:**
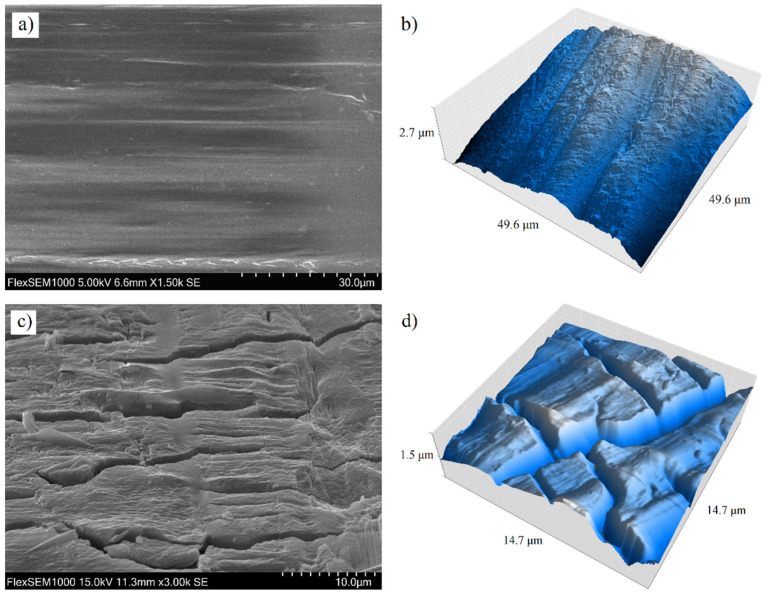
Top surface of a non−degraded stent imaged by (**a**) SEM and (**b**) atomic force microscope. Top surface of a stent after 24 weeks of degradation imaged by (**c**) SEM and (**d**) atomic force microscope.

**Figure 5 materials-14-05462-f005:**
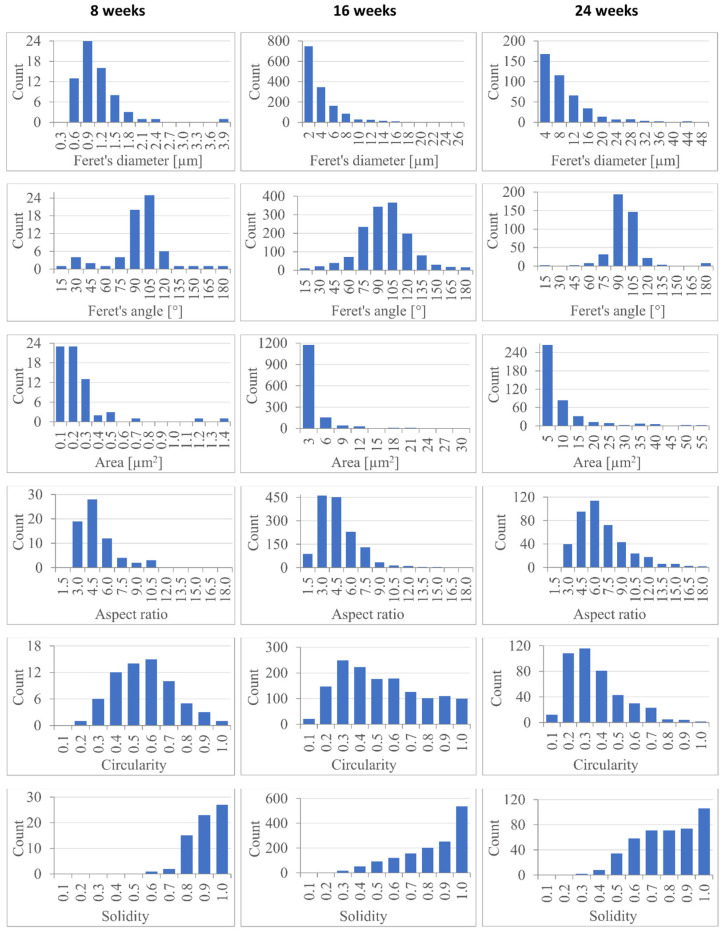
Histograms describing sizes and shapes of cracks observed on the stent surface after 8, 16 and 24 weeks of degradation.

**Figure 6 materials-14-05462-f006:**
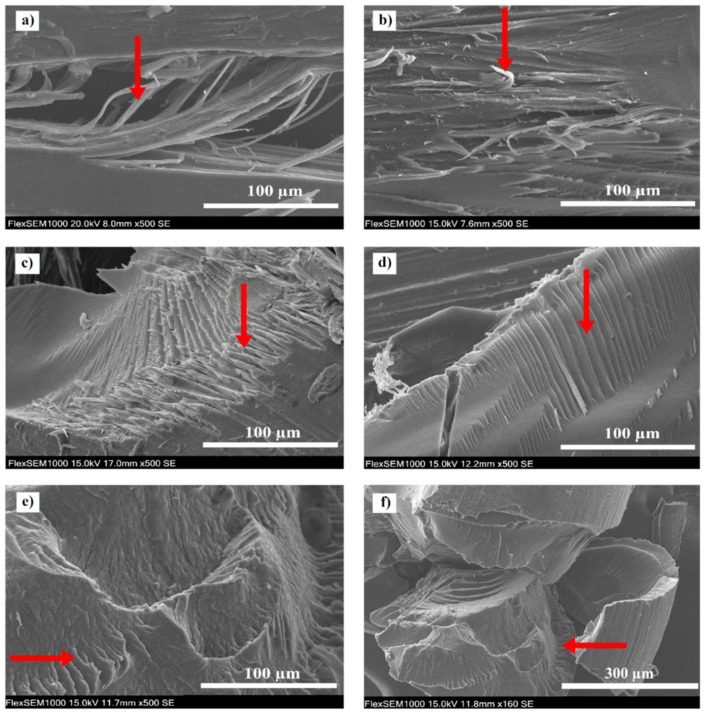
Fracture surfaces of the stents: (**a**) non-degraded, (**b**) 4-weeks-degraded, (**c**) 8-weeks-degraded, (**d**) 16-weeks-degraded and (**e**,**f**) 24-weeks-degraded stent. The arrows point at structures typical of the corresponding level of degradation.

**Figure 7 materials-14-05462-f007:**
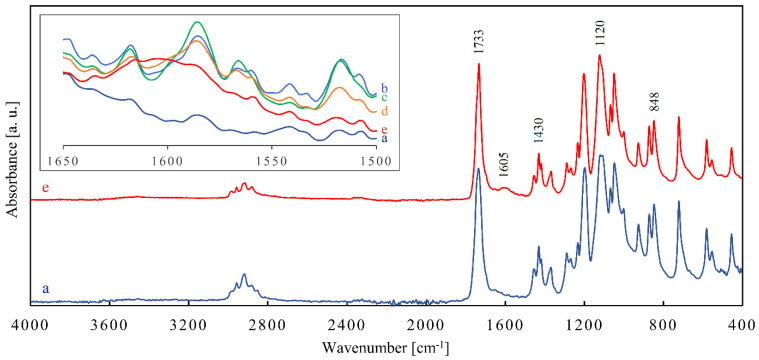
Averaged Fourier transform infrared spectra of the (a) non-degraded stent and (e) 24-weeks-degraded stent. The inset shows small changes in spectra observed as the degradation progressed: (a) non-degraded, (b) 4-weeks-degraded, (c) 8-weeks-degraded, (d) 16-weeks-degraded and (e) 24-weeks-degraded stent.

**Figure 8 materials-14-05462-f008:**
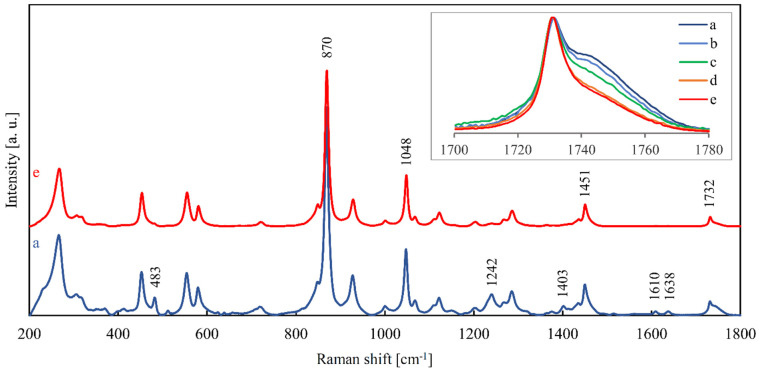
Averaged Raman spectra of the (a) non-degraded stent and (e) 24-weeks-degraded stent after removing the baseline. The inset shows the gradual lessening of the 1732 cm^−1^ peak shoulder progressing with the degradation: (a) non-degraded, (b) 4-weeks-degraded, (c) 8-weeks-degraded, (d) 16-weeks-degraded and (e) 24-weeks-degraded stent. Baseline correction was performed in LabSpec 6 software.

**Figure 9 materials-14-05462-f009:**
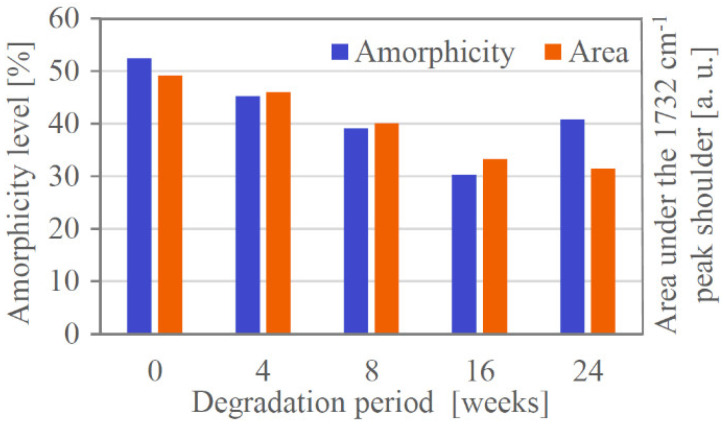
Comparison of the change in PPDX crystallinity (determined by differential scanning calorimetry) and the reduction in the 1732 cm^−1^ peak shoulder.

**Table 1 materials-14-05462-t001:** Crystallinity of the stents depending on the degradation period.

**Degradation Period**	**Crystallinity [%]**
Non-degraded	47.6 ± 2.3
4 weeks degraded	54.8 ± 1.1
8 weeks degraded	60.9 ± 0.7
16 weeks degraded	69.7 ± 2.2
24 weeks degraded	59.2 ± 3.7

**Table 2 materials-14-05462-t002:** Morphological characteristics of the cracks.

**Size/Shape** **Descriptor**	**8 Weeks** **Degraded (Mean)**	**16 Weeks** **Degraded (Mean)**	**24 Weeks** **Degraded (Mean)**	**8 Weeks Degraded (Median)**	**16 Weeks Degraded (Median)**	**24 Weeks Degraded (Median)**
Feret’s diameter [µm]	0.96 ± 0.51	3.04 ± 3.39	7.36 ± 6.96	0.85	1.87	5.21
Feret’s angle [°]	87.3 ± 28.5	89.6 ± 27.0	89.0 ± 19.1	94.5	89.0	87.8
Area [µm^2^]	0.20 ± 0.22	1.97 ± 3.92	6.85 ± 11.86	0.14	0.62	3.00
Aspect ratio	4.00 ± 1.82	3.90 ± 2.24	6.05 ± 2.81	3.61	3.45	5.48
Circularity	0.51 ± 0.17	0.48 ± 0.24	0.32 ± 0.17	0.51	0.45	0.27
Solidity	0.85 ± 0.08	0.77 ± 0.19	0.74 ± 0.18	0.89	0.84	0.76

**Table 3 materials-14-05462-t003:** The dependence of the area under the 1732 cm^−1^ peak shoulder (between 1736 and 1749 cm^−1^) on the degradation period.

**Degradation Period**	**Normalized Area [cm^−1^]**
Non-degraded	9.83 ± 0.22
4 weeks	9.24 (8.98, 9.37)
8 weeks	7.97 (7.80, 8.18)
16 weeks	6.65 ± 0.24
24 weeks	6.30 ± 0.22

## Data Availability

The data presented in this study are available in the Appendix A.

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
