# Peer review of "Raman Spectroscopy as a Novel Method for the Characterization of Polydioxanone Medical Stents Biodegradation"

_materials, 2021, doi:10.3390/ma14185462_

Round 1

Reviewer 1 Report

The manuscript under evaluation deals with application of Raman spectroscopy for the characterization of polydioxanone medical stents biodegradation. This study would be acceptable for publication after revision where authors should address following points:

  1. In the manuscript the word biocompatibility is repeated 3 times. However, biocompatibility is not a property of material but of a material-host system. Therefore, the term “biocompatibility” reflects the ability of a material to perform with an appropriate host response in a specific application. According to Prof. David F. Williams, there is no such thing as a biocompatible material: (http://dx.doi.org/10.1016/j.biomaterials.2014.08.035 ). I would suggest reworking.
  2. More advanced analytical techniques used for evaluation of biodegradation and degradation products structure of similar polymers should be mentioned at the Introduction part of the manuscript (comp. eg.: https://doi.org/10.1021/ma800365m).

Author Response

Dear reviewer. We would like to thank you for your valuable reviews and comments, which, we believe, helped us to improve our manuscript significantly. Our amendments with regard to your requests are listed it the “Reviewer 1.docx” file. Please see the attachment.

Reviewer 2 Report

The manuscript “Raman Spectroscopy as a Novel Method for the Characterization of Polydioxanone Medical Stents Biodegradationauthored by Loskot et al., is very interesting since it addresses new technique to assess degradation behaviour of biodegradable materials for medical applications. However, the authors should answer to some questions and do some modifications before considering this manuscript for publication.

Abstract:

  • The authors are invited to adjust the introductive part of the abstract. They should start with the wide use of PPDX as FDA approved polymer in tissue engineering to talk later about the importance of its degradation evaluation for safety reason.

Introduction:

  • The authors started talking about biodegradable materials including PPDX going through its characterization and properties. Afterwards, they passed to the application of biodegradable materials in Tissue Engineering (TE) i.e. stents. I would favor to start the introduction of the publication by the paragraph in Line 75-91. The authors are invited to move these paragraphs to the beginning of the Introduction Part and then pass through PPDX characteristics and the importance of assessing degradation behaviour of such materials.
  • The authors should add more references concerning the use of PPDX in TE.

Materials and Methods

  • The authors did not provide details about the initial mass of the PPDX stent neither the volume of the PBS in which stents were immersed during degradation studies. The authors are invited to provide the requested information within the new revised manuscript.
  • The authors are invited to specify the statistical tests used for the assessment of stent degradation together with the name of the used software.

Results and discussion

  • The authors are invited to add images for top surface for non-degraded PPDX stents at SEM and AFM analysis as for the 24-week degraded materials.
  • Although the non-degraded stents do not represent cracks on their surfaces, the authors should conduct the same type of analysis including geometrical characteristics on these materials to allow a better comprehension of material behaviour during degradation. A negative control is a necessity.
  • The authors are invited to use arrows within Figure 6 to determine the crack zones allowing a better understanding of stent degradation behaviour and rendering this Figure more explicative.
  • Line 275. Please use the term “asymmetric” instead of “antisymmetric”. Make the correction throughout the text.
  • In the Inset of Figure 7, what are the letter b, c, and d refer to? please explain within the results.
  • The authors should provide the FTIR spectra concerning the analyzed dye.
  • Again, in the Figure 8, what are the letter b, c, and d refer to? Please explain within the results. The authors should also add a spectrum for RS concerning the dye.
  • Please harmonize RS throughout the text since in such paragraph, it is written under both spelling Raman Spectroscopy and RS.
  • The authors should provide more firm results and conclusions concerning what they found interesting using RS to assess degradation. Although the authors put light on the use of RS for degradation evaluation, only short paragraph was addressed to this method. The authors are invited to clarify better the obtained results, to discuss them and conclude with firm conclusions to open new perspectives about the use of RS in such applications.

Author Response

Dear reviewer. We would like to thank you for your valuable reviews and comments, which, we believe, helped us to improve our manuscript significantly. Our amendments with regard to your requests are listed it the “Reviewer 2.docx” file. Please see the attachment.

Reviewer 3 Report

Review report of the manuscript materials-1377456

by Jan Loskot et al.

The authors around Laskot show a novel way to assess the progression of polydioxanone hydrolytic degradation. The investigation has a relevance in the field of the stenting technique, especially for biodegradable stent analysis. The degradation of the stent material was assessed using Raman spectroscopy which is sensitive to monitor the decrease of the dye content in the stents during the degradation.

GENERAL COMMENTS

The work is very interesting and makes a contribution in the field of the stenting technique. The presented study proposes a novel approach for the degradation analysis of biodegradable stent materials which impact in the stenting technique is relevant.

In my opinion the work can be accepted for publication after answering a few minor concerns.

  • More details and introduction on the Raman spectroscopy should be given. Its description and potentiality looks not adequately described.
  • Some references to previous study with this technique should be also given, even with other applications.
  • In general, the Introduction must be imporved adding more references and studies in the field of the stent degradation analysis (even with different techniques).
  • All the variable used for the analysis should be defined and descripted (for instance Feret's variable...)
  • As the stent is commercial, some additional information should be given: is the time of 'complete' degradation predicted?
  • How would the samples size analyzed in the study affect the results (the overall degradation)? Is the size representative of the entire stent? Or in which way is it representative?
  • Even the results are novel and impressive, a section or a paragraph describing the study limitations should be included.

Author Response

Dear reviewer. We would like to thank you for your valuable reviews and comments, which, we believe, helped us to improve our manuscript significantly. Our amendments with regard to your requests are listed it the “Reviewer 3.docx” file. Please see the attachment.

Round 2

Reviewer 2 Report

The manuscript “Raman Spectroscopy as a Novel Method for the Characterization of Polydioxanone Medical Stents Biodegradationauthored by Loskot et al., is now really improved in terms of quality and organization and the authors have amended the requested modifications. The manuscript can be accepted for publications after performing some minor modifications.

Introduction:

  • Line 52-64 should be moved to line 42 so then authors can talk in more detail about PPDX. Please adjust it within the new manuscript.

Results and discussions:

  • In Figure 6, the reviewer appreciates the modifications amended by the authors. However, the authors are invited to reduce the thickness of the arrows within the Figure rendering it clear enough for publication.

Author Response

Dear reviewer. We would like to thank you very much again for your reviews and comments, on the basis of which we modified our manuscript. Our amendments with regard to your requests are listed it the “Reviewer 2_v2.docx” file. Please see the attachment.
